# Altered DNA Methylation Profiles in *SF3B1* Mutated CLL Patients

**DOI:** 10.3390/ijms22179337

**Published:** 2021-08-28

**Authors:** Alicja Pacholewska, Christina Grimm, Carmen D. Herling, Matthias Lienhard, Anja Königs, Bernd Timmermann, Janine Altmüller, Oliver Mücke, Hans Christian Reinhardt, Christoph Plass, Ralf Herwig, Michael Hallek, Michal R. Schweiger

**Affiliations:** 1Institute for Translational Epigenetics, Faculty of Medicine, University Hospital Cologne, 50931 Cologne, Germany; alicja.pacholewska@uk-koeln.de (A.P.); cgrimm6@uni-koeln.de (C.G.); akoeni11@smail.uni-koeln.de (A.K.); 2Center for Molecular Medicine Cologne (CMMC), University of Cologne, 50931 Cologne, Germany; 3Center for Integrated Oncology Aachen Bonn Cologne Duesseldorf, German CLL Study Group, Department I of Internal Medicine, Faculty of Medicine, University Hospital Cologne, 50931 Cologne, Germany; carmen.herling@uk-koeln.de (C.D.H.); Christian.Reinhardt@uk-essen.de (H.C.R.); michael.hallek@uni-koeln.de (M.H.); 4Department of Computational Molecular Biology, Max Planck Institute for Molecular Genetics, 14195 Berlin, Germany; lienhard@molgen.mpg.de (M.L.); herwig@molgen.mpg.de (R.H.); 5Sequencing Core Facility, Max Planck Institute for Molecular Genetics, 14195 Berlin, Germany; timmerma@molgen.mpg.de; 6Cologne Center for Genomics, University of Cologne, 50931 Cologne, Germany; janine.altmueller@uk-koeln.de; 7German Cancer Research Center, Cancer Epigenomics, 69120 Heidelberg, Germany; o.muecke@dkfz.de (O.M.); c.plass@dkfz.de (C.P.); 8German Cancer Consortium (DKTK), 69120 Heidelberg, Germany; 9West German Cancer Center Essen, Department of Hematology and Stem Cell Transplantation, University Hospital Essen, 45147 Essen, Germany

**Keywords:** chronic lymphocytic leukemia, CLL, DNA methylation, *SF3B1* mutation, NOTCH, IKAROS

## Abstract

Mutations in splicing factor genes have a severe impact on the survival of cancer patients. Splicing factor 3b subunit 1 (*SF3B1*) is one of the most frequently mutated genes in chronic lymphocytic leukemia (CLL); patients carrying these mutations have a poor prognosis. Since the splicing machinery and the epigenome are closely interconnected, we investigated whether these alterations may affect the epigenomes of CLL patients. While an overall hypomethylation during CLL carcinogenesis has been observed, the interplay between the epigenetic stage of the originating B cells and *SF3B1* mutations, and the subsequent effect of the mutations on methylation alterations in CLL, have not been investigated. We profiled the genome-wide DNA methylation patterns of 27 CLL patients with and without *SF3B1* mutations and identified local decreases in methylation levels in *SF3B1*^mut^ CLL patients at 67 genomic regions, mostly in proximity to telomeric regions. These differentially methylated regions (DMRs) were enriched in gene bodies of cancer-related signaling genes, e.g., *NOTCH1*, *HTRA3*, and *BCL9L*. In our study, *SF3B1* mutations exclusively emerged in two out of three epigenetic stages of the originating B cells. However, not all the DMRs could be associated with the methylation programming of B cells during development, suggesting that mutations in *SF3B1* cause additional epigenetic aberrations during carcinogenesis.

## 1. Introduction

Chronic lymphocytic leukemia (CLL) is the most common leukemia in the Western world and mainly affects elderly people [1]. Although the CLL phenotype is quite specific and homogenous, the clinical outcome is extremely heterogeneous [1]. The clinical outcome is partly associated with the mutational status of the immunoglobulin heavy chain variable region (*IGHV*) as patients with a high level of somatic mutations in IGHV (M-CLL) have a better prognosis than patients with no or a low level of somatic mutations in this region (U-CLL) [1,2,3].

Through the development of new high-throughput sequencing technologies, additional genomic alterations have been identified, which are associated with poor prognosis or insufficient therapy response. The strongest impact has been found for *del(17p)* and mutations in tumor protein P53 (*TP53*). Additionally, shorter progression-free survival is conferred by mutations in the splicing factor 3b subunit 1 (*SF3B1*), ATM serine/threonine kinase (*ATM*), ribosomal protein 15 (*RPS15*), and Notch receptor 1 (*NOTCH1*) [4,5,6,7,8,9,10]. Mechanistic insight into how these genomic alterations lead to poor prognosis or therapy resistance in CLL is largely missing. One of the most frequently mutated genes in CLL is *SF3B1*, encoding a component of the splicing machinery. Patients with *SF3B1* mutations (*SF3B1*^mut^) have a poor prognosis [5] and *SF3B1* alterations are associated with chemotherapy refractory disease [7]. *SF3B1* mutations cluster in the HEAT repeat domain (Huntingtin, Elongation factor 3—EF3), protein phosphatase 2A (PP2A), and the yeast kinase TOR1) with the most common mutation in CLL being *p.K700E* [5,9]. Depletion of *SF3B1* transcripts by small interfering RNA (siRNA) in cell culture leads to an alteration of exon usage, in most cases causing a decrease in cassette exon inclusion [11]. Long read sequencing also revealed differential 3′ splice site changes and a strong downregulation of intron retention events associated with *SF3B1* mutations [12]. Recently, it has been shown that blood malignancies, in particular, such as leukemias, have a strong link between mutations in splicing factors and epigenetic dysregulation [13]. As such, SF3B1 interacts with the chromatin remodeling complex WICH (WSTF-SNF2H) [14] and with the polycomb group proteins: polycomb group ring finger 2 (PCGF2) and ring finger protein 2 (RNF2) [15]. A direct interaction between SF3B1 and polycomb repressive complex 2 (PRC2) was shown in mice. As such *Sf3b1*^+/−^ mice exhibited a similar phenotype as *PcG*^mut^ mice, i.e., various skeletal alterations along the anterior-posterior axis [15]. Moreover, hypermethylation of polycomb-repressed regions was observed in the proliferating fraction of circulating CLL [16]. Furthermore, SF3B1 interacts with nucleosomes in an RNA-independent manner and is preferentially associated with GC-rich exons [11]. However, a detailed characterization of epigenomic changes associated with *SF3B1*^mut^ CLL is still largely missing.

Epigenomic alterations are emerging as powerful prognostic indicators in CLL [17,18]. It has been found that CLL genomes, when compared to normal B cells, are globally hypomethylated [19,20,21] and that M- and U-CLL classes show distinct methylomes [22]. Hypomethylations are found at gene bodies [23], especially at repetitive sequences, such as Alu, long interspersed nuclear elements-1 (LINE-1), and satellite-α (SAT-α) repeats [24]. Epigenetic profiling using DNA methylation arrays identified three subgroups of CLL, reflecting the developmental stage of the B cells from which the CLL cells originated [23,25]. Normal B cell maturation from naive to high-maturity memory B cells is accompanied by unidirectional DNA methylation changes, of which most show a decrease in DNA methylation. Such epigenetic changes during differentiation processes are also referred to as epigenetic programming. Epigenetic-defined CLL subgroups were therefore named low-programmed CLL (LP-CLL), intermediate-programmed CLL (IP-CLL), and high-programmed CLL (HP-CLL) [25]. LP-CLL is enriched for unmutated *IGHV* (U-CLL) and is associated with a poor prognosis, whereas HP-CLL is enriched for mutated *IGHV* (M-CLL) [23].

Recent studies have revealed wide-spread intratumor heterogeneity of the methylation in CLL [18,26,27]. Upon maintenance of DNA methylation, the methylation of one CpG is influenced by neighboring CpGs, yielding concordant methylation states [28]. In contrast, discordant methylation states are associated with active reprogramming. In cancer, a higher degree of discordant neighboring CpG methylation at promoter sites has been associated with worse prognosis [26]. According to this study, the median time for ‘failure free survival’ (FFS), meaning the time between the first and the second treatment or death, decreases from 44 months to 16.5 months for patients with a high proportion of discordant methylation in the promoter. This suggests that DNA methylation is a predictor for prognosis, but whether it is causative or just a confounding factor to the malignant process is so far unknown. Along this line, it is unclear whether DNA methylation and oncogenic mutations are independent prognostic factors or if are functionally related. Landau et al. observed that the presence of sub-clonal drivers overruled the increased risk associated with the elevated PDR (proportion of discordant reads), suggesting that either the heterogeneous methylation facilitates mutational processes, or that the mutations exert their functions through epigenetic mechanisms [26].

Moreover, all three DNA methyltransferases (DNMTs) are subjected to alternative splicing [29,30,31] and, therefore, mutations in splicing factors can potentially lead to changes in the methylation profiles, either directly by affecting the splicing of the DNA methyltransferases and DNA demethylases genes, or by alternatively spliced isoforms of the chromatin and methylation regulators, such as non-coding RNAs [32]. Thus, to achieve a better understanding on the methylome changes in CLL patients and how they might be connected to mutations in *SF3B1*, we compared genome-wide methylation profiles of CLL patients with (*SF3B1*^mut^) and without (*SF3B1*^WT^) *SF3B1* mutations.

## 2. Results

### 2.1. CLL Patients with SF3B1 Mutations Feature Local Hypomethylations

Although overall hypomethylation in CLL patients compared to normal B cells is known [23,24], the impact of the *SF3B1* mutations on the epigenome is unclear. We used DNA of 27 patients with (*SF3B1*^mut^, *n* = 13) and without (*SF3B1*^WT^, *n* = 14) *SF3B1* mutations (Appendix A), and investigated genome-wide methylation profiles using the methylated DNA immunoprecipitation sequencing (MeDIP-seq) technology. Although principal component analysis with the 1,000 most variable regions within CG islands (CGIs) revealed that samples clustered according to the *IGHV* mutational status, and some separation could also be observed based on the *SF3B1* mutational status, no clear clusters grouped by the *SF3B1* genotype were visible (Figure 1A). This suggested that *SF3B1*^mut^ has no widespread effect on the methylation patterns of the CLL patients. The distribution of sex or age between the *SF3B1*^WT^ and *SF3B1*^mut^ patients was not significantly different (Mann–Whitney test *p*-value = 0.94) (Appendix A). Using the QSEA package [33], we identified 67 significantly hypomethylated, but no hypermethylated regions (adjusted *p*-value (false discovery rate, FDR) < 0.05, and |log2(fold change)| ≥ 1) (Appendix A). We validated 16 of the significantly differentially methylated regions (DMRs) with the EpiTYPER (Appendix A) and observed a significant correlation (R = 0.71, *p*-value < 2.2 × 10^−16^) between the methylation levels of the regions estimated with MeDIP-seq/QSEA and EpiTYPER (Figure 1B,C and Appendix A). Although the EpiTYPER data cannot provide methylation information for individual CpGs localized on the same DNA strand, the bulk methylation level of neighboring CpG sites at 14 DMRs with more than one CpG tested seems mostly concordant (Appendix A). A hierarchical clustering using the beta normalized methylation values at the 67 DMRs clearly separated *SF3B1*^mut^ from *SF3B1*^WT^ (Figure 2A).

We next compared our results to published methylation data created with methylation arrays that, even though they did not focus on the *SF3B1*^mut^ effect, did contain methylation data of *SF3B1^mut^* patients. Kulis et al. used 139 CLL patients, 8 of which with *SF3B1* mutation [23]. The authors identified 64 hypo- and 30 hypermethylated differentially methylated CpGs between *SF3B1*^mut^ and *SF3B1*^WT^. Although 74 of these 94 CpGs (79%) were within genomic regions sufficiently covered by sequencing reads in our data and tested for differential methylation, none of the DMRs identified overlapped the exact differentially methylated CpGs identified by Kulis et al. [23]. However, we identified DMRs within three genes that contained a differentially methylated CpG reported by Kulis et al. [23]: *BCL9L*, *MYB*, and *NCOR2*. The CpGs and DMRs within these genes had the same direction of the methylation change; they were hypomethylated in CLL with *SF3B1*^mut^ compared to CLL with *SF3B1*^WT^. We next used an even larger dataset, one that encompasses the dataset from Kulis et al. [23], available from BloodCancerMultiOmics2017 R package [34]. In total, 174 CLL patients with known *SF3B1* mutational status (148 *SF3B1*^WT^ and 26 *SF3B1*^mut^) were analyzed for 435,102 CpGs (all CpG sites with a single nucleotide polymorphism, SNP, were removed by the authors). We used the limma R package [35] to analyze the array data sets and found that only 18 from our 67 DMRs (27%) overlapped at least one CpG (in total, 27 CpGs) included in the methylation array dataset, suggesting that the differential methylation might be located outside of regions captured with the array technology. The methylation values of the array data within all of these 18 regions were altered in the same direction as we have identified it within our dataset. Apart from one CpG with no change in methylation, the CpGs overlapping these DMRs showed a slight hypomethylation in CLL *SF3B1*^mut^ patients with a maximum |log2(fold change)| = 0.3, within the *ACOX3* gene (Appendix A, column AI). This again puts our results in line with previously published data. Among the 27 CpGs overlapping the 18 DMRs, there were CpGs already reported by Wierzbińska as CLL-specific (*n* = 8), or B cell-specific (*n* = 8) [36] (Appendix A). The mean methylation values for CLL *SF3B1*^WT^ and CLL *SF3B1*^mut^ of the 18 CpGs were significantly correlated between the array and the MeDIP-seq based data (Pearson R = 0.47, *p*-value = 0.0038) (Appendix A).

### 2.2. Hypomethylations in CLL Patients with SF3B1 Mutation Are Enriched in Gene Bodies and Subtelomeric Regions

We next asked if the changes in DNA methylation in *SF3B1*^mut^ samples are enriched in certain genomic regions or if they are evenly distributed across the genome. While we identified DMRs on almost all chromosomes, except chromosomes 14 and 18 (Figure 2B,C), the density of DMRs for the chromosomes varied (chi-squared test *p*-value = 0.00015). We observed the highest number of DMRs on chromosomes 9 and 19 (Figure 2B,C) (Appendix A). Chromosome 19 is known for its high density of genes and CG content [37]. We therefore corrected the analysis for the CGI content and identified the largest enrichment of DMRs on chromosome 9 (Appendix A). We did not observe any major deletions or insertions on chromosomes 9 or 19 based on the QSEA estimation (Appendix A). All eight DMRs identified on chromosome 9 (as well as many DMRs on other chromosomes) were located within 15 Mbp from the chromosome end in proximity of telomeric regions (Figure 2C) [38]. In fact, 43 of our 67 DMRs (64%) were located within 10 Mbp from the chromosomal start/end, and 28 of these (42% total) were located even more peripheral, within 5Mbp from the chromosomal ends. This result suggested that the hypomethylation may involve spatially and therefore possibly functionally related chromosomal regions. The subtelomeric DMRs on chr9 overlapped gene bodies of *EHMT1*, *NOTCH1*, *VAV2*, *PRRC2B*, *NEK6*, and the promoter region of *LCN10*.

Furthermore, to examine the distribution of DMRs in the genomic context, we annotated the DMRs and counted how many DMRs spanned different genomic features (Figure 3A). DMRs were significantly enriched in gene bodies (Figure 3B) with more than half (*n* = 48, 72%) of the DMRs located within gene bodies, mostly intronic regions (*n* = 40). Furthermore, we observed that DMRs were enriched in transcription factor binding sites (TFBS, *n* = 29, 43%). These TFBS were mostly downstream to gene promoters, except six, which were located in the promoters of *TGFBR3*, *RGPD8*, *HTRA3*, *LCN10*, *FAM174B*, and *IL17C* (Appendix A). Of those, four had a CpG island annotated within the promoter region (*RGPD8*, *HTRA3*, *FAM174B*, *IL17C*). We next performed TFBS enrichment analysis of the 122 transcription factors (TFs) included in the ENCODE Uniform TFBS track [39] derived from ChIP-seq experiments [40,41,42]. We identified IKZF1 (IKAROS) and BHLHE40 as the most significantly enriched transcription factors with 12 and 8 hypomethylated DMRs at their binding sites, respectively (Figure 3C). Both TFs are critical for B cell development [43,44]. This is particularly interesting in regard to the question if the detected differential methylations are due to different B cell developmental stages where the *SF3B1*^mut^ cases develop from. However, the highest odds ratio was identified for the histone deacetylase HDAC6 with two DMRs at the promoter regions of *HTRA3* and *FAM174B*. HDACs’ role in the initiation and progression of cancer has been extensively studied, as reviewed in [45]. Among chromatin states, we found weak and strong enhancers as predominant sites for DNA hypomethylations (Figure 3D). Among the 29 DMRs within enhancer regions, 21 (72%) were within 1 Mb from a promotor of a gene with significantly different expression levels (FDR < 0.05, no threshold on log2(fold change, Appendix A), e.g., *XXYLT1*, *HTRA3*, and *ARID3A* genes. Eleven DMRs (16%) were located at ten unique promoter regions (*BCL9L*, *HTRA3*, *HPCAL1*, *IL17C*, *RGPD8*, *NCOR2*, *LCN10*, *AC107959.1*, *FAM1748*, *TGFBR3*) and two genes (*HTRA3* and *FAM1748*) had higher expression and hypomethylated promoters in CLL patients with *SF3B1*^mut^ compared to *SF3B1*^WT^ (Appendix A).

Subsequently, to gain more biological insight into the differential methylations in CLL patients with *SF3B1* mutation, we used all 40 genes containing at least one DMR within their gene body or promoter region and subjected the list to a gProfiler functional enrichment analysis. This analysis identified the NOTCH signaling pathway (KEGG pathway, g:SCS adjusted [46] *p*-value = 1.49 × 10^−2^) containing three of the genes (*DTX1, NCOR2, NOTCH1*) as significantly affected by differential methylations (Appendix A).

Motif enrichment analysis of the 67 regions did not reveal any significantly enriched transcription factor motif after multiple testing correction (Appendix A).

### 2.3. SF3B1^mut^ Is Associated with Aberrant Methylation and Is Partially Related to the Developmental B Cell Epigenetic State

It has been previously shown that the methylation profile in CLL reflects, besides tumor specific alterations, the developmental state of the B cells from which the tumor has derived [25]. Accordingly, CLL cases have been classified into three stages of B cell development: low-programmed (LP-CLL), intermediate-programmed (IP-CLL), and high-programmed (HP-CLL) [25]. This classification is based on clustering of methylation levels from regions containing binding sites for transcription factors involved in B cell development (AP-1, EBF1, RUNX3) and transcriptional elongation [25]. The authors identified 18 regions, which sufficiently separate the three B cell developmental stages [25]. Using beta-transformed methylation levels calculated by the QSEA software [33] for these 18 regions, we clustered our 27 samples together with the 329 CLL Research Consortium samples analyzed by Oakes et al. [25] (Figure 4A and Appendix A).

The occurrence of *SF3B1* mutations have been associated with less differentiated states of B cells with the *SF3B1*^mut^ CLL resembling more naïve B cells whereas *SF3B1*^WT^ CLL resembling more memory B cells based on their methylation profiles [51,52,53]. We therefore tested to what extent the *SF3B1*^mut^-associated hypomethylated regions can be explained by the developmental stage, and to what extend they can be attributed to the effects of the *SF3B1*^mut^ during carcinogenesis. While four of the *SF3B1*^WT^ samples clustered within the HP-CLL cluster, we did not observe any sample with a *SF3B1* mutation in this cluster (Figure 4A,B). This was confirmed by independent clustering of our 27 samples, as all HP-CLL samples (19, 21, 22, 36), none of which carried *SF3B1*^mut^, created a separate cluster (Figure 1A and Appendix A).

In order to evaluate if the identified 67 DMRs between *SF3B1*^mut^ and *SF3B1*^WT^ were due to different developmental stages of the originating B cells, we looked at the methylation differences based on beta-normalized methylation values at the 67 regions among the developmental stages represented by LP-, IP-, and HP-CLL subtypes (Figure 4C). We observed that, for *SF3B1*^WT^ CLL, the methylation levels at these regions significantly increased between the LP- and IP-CLL stages, whereas *SF3B1*^mut^ CLL samples showed stable methylation levels, which were significantly lower compared to the corresponding stage in the *SF3B1*^WT^ CLL (Figure 4C). At 22 from the 67 DMRs (33%) identified between *SF3B1*^mut^ CLL and *SF3B1*^WT^ CLL samples the methylation level changed (≥20%) between LP- and IP-CLL among the *SF3B1*^WT^ CLL samples (Figure 4D, Appendix A), suggesting that these DMRs are involved in the normal B cell developmental process.

These 22 B cell development-related DMRs are located in 16 genes, including genes involved in B cell-specific functionality, epigenetic remodeling, and carcinogenesis, such as *TGFBR3*, *EHMT1*, *ACOX3*, *SEPTIN9*, *MYB*, *FAM174B*, *ARID3A*, *XXYLT1*, and *RHOBTB2.* Although we observed ≥ 20% change in the methylation at the 22 of 67 DMRs between LP-CLL and IP-CLL among the *SF3B1*^WT^ CLL samples, the difference was lost in *SF3B1*^mut^ (Figure 4C). An additional 12 of our DMRs showed methylation differences (≥20%) when we compared HP-CLL to IP-CLL among the *SF3B1*^WT^ CLL samples, and 33 of the 67 DMRs (49%) showed stable methylation profiles among the *SF3B1*^WT^ CLL samples (LP-CLL vs. IP-CLL and IP-CLL vs. HP-CLL, Appendix A), indicating that about half of the methylation changes observed between *SF3B1*^WT^ and *SF3B1*^mut^ are independent of B cell developmental stages.

Furthermore, we compared our list of DMRs with the 10,000 regions from Oakes et al. and concluded that most significantly change their methylation levels during physiological B cell maturation [25]. Only 4 of our 67 DMRs (6%) overlapped the 10,000 regions associated with B cell development reported by the authors [25] (Appendix A). It is worth noting that all four overlapping DMRs, including DMRs within *BCL9L* and *NOTCH1*, were hypermethylated in high-maturity memory B cells and hypomethylated when comparing *SF3B1^mut^* vs. *SF3B1^WT^* CLL, suggesting an association of *SF3B1* mutations with less mature B cell developmental stages in CLL.

In addition, to further investigate if the observed DMRs between *SF3B1*^mut^ and *SF3B1*^WT^ were related to the B cell maturation or not, we compared our list of DMRs with epigenetic B cell programming sites identified using a methylome-based cell-of-origin modelling framework [36]. The authors identified linear dynamics of the methylation changes at 59,329 CpGs occurring during normal B cell development across six B cell differentiation stages, from naive to memory B cells (B cell-specific sites). CpGs with deviations from the expected methylation levels in CLL (CLL-specific) were classified into four classes: B cell-specific developmental sites hypomethylated (class A, *n* = 5757) and hypermethylated (class B, *n* = 183); and non B cell-specific sites hypomethylated in CLL (class C, *n* = 4238) and hypermethylation in CLL (class D, *n* = 157). The CLL-specific CpGs are expected to be associated with the tumorigenic transformation to CLL [36]. Eight (12%) of our DMRs overlapped the B cell-specific developmental sites reported by the authors [36], and seven (10%) overlapped CLL-specific CpGs not related to the B cell differentiation program (Appendix A). This indicates that a part of the differential methylations detected in *SF3B1*^mut^ compared to *SF3B1*^WT^ patients is related to the normal B cell differentiation process, and that the other part is specific to *SF3B1*^mut^ CLL.

## 3. Discussion

Methylation is known to regulate splicing [54,55,56,57], but alternative isoforms of DNA methyltransferases genes or genes regulating chromatin conformation or methylation can also modulate methylation profiles [32]. Although mutations in genes required for splicing and methylation commonly occur in leukemia, and a mutation in a splicing factor (*SRSF2*) has been shown to impact methylation in acute myeloid leukaemia [58], the interaction of these two processes has not been described in CLL patients carrying *SF3B1*^mut^. To acquire additional insight into the methylome differences in CLL patients with and without *SF3B1* mutations, we analyzed genome-wide methylation profiles using MeDIP-seq. We identified 67 regions with significantly lower methylation levels in *SF3B1*^mut^ CLL (Figure 2A) which we partly validated with the EpiTYPER assay.

The question remains what the cause for the altered methylation pattern might be. So far, there are no reports on the *SF3B1* mutation causing differentially spliced isoforms of DNMTs in CLL patients, which is also in line with our data. However, altered splice variants in euchromatic histone lysine methyltransferase 1 (*EHMT1*) have been significantly associated with *SF3B1*^mut^ CLL patients [59], and were also detected by us to contain a hypomethylated region (Appendix A). In fact, the *EHMT1* and *UCKL1* genes were the only genes that overlapped the DMRs reported here, and were listed as genes with altered splicing associated with *SF3B1* mutation in CLL patients [59]. Although most histone methyltransferases are independent from DNA methylation, they are involved in gene repression [60] and DNA damage [61]. Furthermore, *EHMT1*, in particular, has also been associated with DNA methylation [62,63] and, therefore, this enzyme requires further investigation in *SF3B1*^mut^ CLL patients.

The methylation changes observed seemed to be only partially affected by the *IGHV* mutational status and allowed to clearly separate the samples by the *SF3B1* mutational status (Figure 2A). The hypomethylated regions were distributed across the genome; however, chromosomes 9 and 19 showed higher numbers of DMRs with many DMRs located close to telomeric regions (Figure 2C). The hypomethylation of regions in CLL compared to normal B cells [19,24] and further hypomethylation of CLL patients with *SF3B1*^mut^ may have a role in the worse prognosis of these patients. Interestingly, all eight DMRs on chr9, which contained, for example, a DMR within the *NOTCH1* gene, as well as many DMRs on other chromosomes, were located in a close proximity to telomeric regions. Short telomers have been already associated with anomalies in *SF3B1* showing worse prognosis in CLL patients [59,64,65]. However, the potential link between differential methylation close to the subtelomeric region and shorter telomeres in CLL patients with *SF3B1*^mut^ requires further studies. We also observed hypomethylation in two other genes involved in NOTCH signaling: *DTX1* and *NCOR2*. Mutations in the *NOTCH1* gene are frequent in CLL [4,5,6,7,8,9,10] and NOTCH signaling was associated with CLL progression [66,67,68]. In addition, NOTCH1 pathway was shown to be activated in CLL patients with *SF3B1* mutation [59,69].

In agreement with previous studies, most of our DMRs were located in genic regions (Figure 3B). The genes with DMRs were significantly enriched within the NOTCH signaling pathway (Appendix A). Interestingly, 12 DMRs included a binding site for the IKAROS (IKZF1) transcription factor (TF). These 12 DMRs were associated with eight genes: *ACOX3*, *ADAT3*, *ARID3A*, *FAM174B*, *NOTCH1*, *RGPD8*, *SCAMP4*, and *XXYLT1*. IKAROS is involved in B cell development [59] and has previously been implicated with CLL [70]. IKAROS expression increases during B cell differentiation and half of all genes upregulated during B cell development are IKAROS targets [71]. Moreover, IKAROS proteins are destructed by lenalidomide [72,73], a drug shown to act on CLL cells in vitro [74] and tested in the CLLM1 trial, where it improved progression-free survival. However, since a subset of treated patients developed B cell acute lymphoblastic leukemia, lenalidomide treatment was terminated in the CLLM1 study [75]. Of note, none of the patients analyzed in this study were treated with lenalidomide. Not surprisingly, 9 of the 12 DMRs within IKAROS binding sites were in regions identified as associated with B cell development and reported here or in previous studies [25,36]. The differential methylation between *SF3B1*^mut^ and *SF3B1*^WT^ CLL patients may, at least in part, be influenced by the differentiation state of the originating B cells [51,52,53]. In agreement with this hypothesis, all of our 14 *SF3B1*^mut^ samples were classified as early- (LP-CLL) and intermediate- (IP-CLL) programmed CLLs, and the role of IKAROS in B cell development has been highlighted in the early stages [43,76,77,78]. However, about half of the DMRs reported here seem to be independent from the B cell differentiation process.

Induction of IKAROS in CLL cells is associated with poor disease outcome [79] and promotes BCR signaling [80]. Taking into account that IKAROS tumor suppressive capacity includes an induction of enhancers in T cells [81], the hypomethylation in *SF3B1*^mut^ patients at IKAROS binding sites and the enrichment of the hypomethylation sites in weak enhancers is noticeable and requires further investigation.

IKAROS family proteins interact with nucleosome remodeling and deacetylase (NuRD) and PRC2 [82]. In T cells, IKAROS interacts with PRC2, thereby mediating epigenetic repression at stem cell-associated genes [83]. Such an interaction of IKAROS with PRC2 links IKAROS to DNA methylation regulation. Although binding of IKAROS to PRC2 was not observed in B cells, a loss of IKAROS function results in ectopic enhancer activation accompanied by a loss of the PRC2-mediated repressive histone modification H3K27me3 in the corresponding promoter regions [84]. In addition, the IKAROS family member, IKZF3 (AIOLOS), is recurrently mutated in CLL, with an incidence of 2% carrying the hotspot mutation *p*.L162R [5], emphasizing the importance of this transcription factor family for lymphoid malignancies.

Although significant, but with only two DMRs in TFBSs, is the histone deacetylase, HDAC6. The hypomethylated two sites for this region were located within promoter regions *HTRA3* and *FAM174B*. Although *FAM174B* has binding sites for more TFs, *HTRA3* has binding sites in this region only for HDAC6 and the PRC2 component, SUZ12. Histone deacetylase inhibitors have been already in phase II clinical trials to treat patients with breast cancer [85]. Furthermore, 30% of the DMRs had a TCF7L2 binding motif (Appendix A), a key player in Wnt signaling [86,87,88]. A DMR was also identified in the *BCL9L* gene—an activator of Wnt signaling associated with B cell malignancies that have been implicated in cancer development and epithelial-mesenchymal transition through a down-regulation of c-Myc, cyclin D1, CD44, and vascular growth factor in tumor cells [89]. Hypomethylation within this gene in CLL patients with *SF3B1* mutation has been previously reported [23] and it has been recently shown that BCL9 and BCL9L promote tumorigenicity in a triple negative breast cancer mouse model through immune-dependent (TGF-β) and immune-independent (Wnt) pathways [90]. Interestingly, 25 of the 67 DMRs (34%) contained a TCF7L2 TF motif, and transcription factor 7-like 2 plays a key role in the Wnt signaling pathway [86,87,88]. A potential role of the *BCL9L* hypomethylation in poor prognosis of the CLL patients with *SF3B1*^mut^ should be further investigated.

In comparison with previous studies, our MeDIP-seq approach covers a broader part of the genome. For example, 450-K methylome arrays analyze 482,486 CpGs, most of which are located in genic regions and CpG islands [91]. Since the human genome contains roughly 28 million CpGs, around 1.6% of all CpGs are amenable by 450-K arrays. In contrast, MeDIP-seq accesses the mappable genome with a CpG content of at least 3% [92] at a resolution of approximately 250 bp. Only 435,102 CpGs from the 450-K methylation array [34] were amenable for the differential methylation testing, whereas MeDIP-seq analyzed by QSEA allowed for testing of 6,540,448 250 bp windows, covering 335,033 CpGs (77%) from the 450-K methylation array. This difference in coverage may also explain the small overlap of our results with previous studies. The larger fraction of the epigenome analyzed our study added new insight into the understanding of the disease.

It has been previously suggested that methylation differences identified among CLL samples may derive from a different maturation status of the B cells at the time of tumorigenesis [25,36]. We therefore classified our patients to the three CLL subtypes as defined previously [25]. Similarly to other studies, *SF3B1*^mut^ CLL samples were classified as LP- and IP-CLL subtypes [51,52,53]. Even though this dependency is not significant in our data due to a low number of samples (chi-squared test *p*-value = 0.09), it is in agreement with other studies, which reported the highest occurrence of CLL with *SF3B1*^mut^ in naïve B cell-like CLL (LP-CLL), and lowest in memory B cell-like CLL (HP-CLL) [51,52,53]. In line with this, we observed an intersection between DMRs reported here and regions differentially methylated during physiological B cell maturation [25,36], indicating that *SF3B1* mutations are, at least in part, associated with B cell developmental stages (Appendix A). Some of them have been already associated with CLL-specific methylation changes [36]. However, there remains a large fraction of DMRs not related to B cell development, indicating that these regions are associated with *SF3B1* specific functions. How *SF3B1*^mut^ and changes in the methylome are exactly related needs further investigation. So far, we have shown that CLLs carrying *SF3B1*^mut^ contain differences in their DNA methylation patterns and that these changes affect genes involved in BCL9 and NOTCH signaling, among other processes. Thus, our findings provide a rich insight for further studies of the causes and consequences of *SF3B1*^mut^ induced changes in gene expression. This might, in the long term, provide the basis for the development of new therapeutic options.

## 4. Materials and Methods

### 4.1. Sample Preparation

Clinical information of the patients is summarized in Appendix A. Staging was performed, according to Binet et al. [93], using blood cell counts. Patients were classified into Binet stage C when patients were anemic (hemoglobin < 10 g/mL) and/or displayed thrombocytopenia (thrombocytes < 100,000/µL), and into stage A/B when patients had more hemoglobin or thrombocytes. One patient had exactly 100,000 thrombocytes/µL and was therefore staged B/C (Appendix A). *TP53*, *SF3B1*, *ATM*, *XPO1*, and *NOTCH1* mutational status were analyzed by a PCR panel followed by next-generation sequencing, as described in [94]. In particular, the complete coding region for *SF3B1*, *TP53*, and *ATM* was analyzed, and *XPO1* exons 12, 13, and 15 were evaluated. *IGHV* mutational status was determined, as previously described [95]. Peripheral blood B cells were isolated via negative selection using RosetteSep immunodensity-based cell separation (Stemcell Technologies, Vancouver, BC, Canada). The cell purity of CLL B cells was analyzed by flow cytometry, and cells co-expressing CD5/CD19 were ≥ 90%. DNA was isolated from frozen B cells using the QIAamp DNA Mini Kit (Qiagen, Hilden, Germany) or obtained from the local biobank. Informed consent was obtained from all patients and the study was approved by the local ethics committee. RNA was isolated using TRIzol (Thermo Fisher Scientific, Waltham, MA, USA) according to the manufacturer’s instructions. A DNAse digest was performed using 2 µg of RNA, and the quality of the RNA was assessed using a Bioanalyzer (Agilent Technologies, Waldbronn, Germany).

### 4.2. Methylated DNA Immunoprecipitation Sequencing (MeDIP-Seq)

The integrity of DNA was evaluated on a 1% agarose gel and 1.3 µg was subjected to MeDIP. Methylated DNA immunoprecipitation (MeDIP) was based on a method developed by [96] with modifications. In brief, 1.3 µg of DNA in 65 µL of TE was sheared to a size of 100–300 bp using a Covaris M220 (Covaris Ltd., Brighton, UK). The size was controlled on a 1% agarose gel. Library preparation was performed using the TruSeq DNA sample preparation kit (Illumina, San Diego, CA, USA) and unmethylated TruSeq indexed adaptors. Library preparation reactions were purified using AMPure XP beads (Beckman Coulter GmbH, Krefeld, Germany) and the adapter-ligated DNA was denatured at 95 °C for 10 min and subjected to MeDIP. For the MeDIP reaction, 5 μg of the monoclonal antibody clone 33D3 directed against 5-methylcytidine (Eurogentec GmbH, Cologne, Germany) was coupled over night at 4 °C to magnetic Dynabeads M-280 sheep anti-mouse IgG (Thermo Fisher Scientific, Waltham, MA, USA). Subsequently, denatured DNA and antibody coupled Dynabeads were incubated at 4 °C for 4 h in immunoprecipitation buffer (IP: 10 mM sodium phosphate buffer (pH 7.0), 140 mM NaCl, 0.25% Triton X100) followed by three washes with IP buffer. DNA was eluted from the beads in 50 mM of Tris-HCl (pH 7.5), 10 mM of EDTA, and 1% SDS at 65 °C for 15 min. The eluted DNA was diluted 1 to 1 with 10 mM of Tris at pH 8.0 and 1 mM of EDTA, and treated with proteinase K (0.2 µg/µL) for 2 h at 55 °C. The methylated DNA was purified using the QIAquick PCR Purification Kit (Qiagen, Hilden, Germany). Following MeDIP enrichment, libraries were amplified using 10 PCR-cycles, size-selected using an agarose gel, and purified DNA was quantified using the Quant-iT dsDNA HS Assay Kit and a Qubit 1.0 Fluorometer (Thermo Fisher Scientific, Waltham, MA, USA). Next, 50 bp single-end reads were generated on a HiSeq4000 (Illumina, San Diego, CA, USA).

### 4.3. Sequencing Reads Processing

MeDIP sequencing reads were aligned to the hg38 reference genome (Genome Reference Consortium GRCh38) with BWA v0.7.15 aln followed by samse modules [97]. Patients’ sequencing data are available from the corresponding author upon reasonable request.

### 4.4. Differential Methylation Analysis

MeDIP-seq reads were processed in R with QSEA v.1.14.0 [33], according to the package recommendations. In brief, reads where counted per genomic 250 base window, and CpG enrichment profiles were calibrated based on highly methylated genomic regions in 196 primary CLL samples of the PACE project, as retrieved from the Bioconductor package CLLmethylation [34] (methylation > 80% in at least 95% of the samples). Differentially methylated regions (DMRs) were called with the implemented likelihood ratio test, based on generalized linear models, and *p*-values were corrected for multiple testing by false discovery rate (FDR) [98]. We considered a region to be differentially methylated if the FDR was smaller than 0.05 and |log2 fold change| ≥ 1. Moreover, we excluded regions with expected CpG density below 4 CpGs per sequencing fragment and all fragments from X and Y chromosomes. DMRs were annotated with BSgenome v.1.56.0. and RefSeq release 71, ENCODE Encyclopedia v.3 as of 24th April 2014, enhancer data from [48], and CGI as described in [47], transcription factor binding sites (TFBS) from ENCODE Uniform TFBS track [39,40,41,42]. Chromatin states coordinates for GM12878 cell line [49,50] were converted from hg19 to hg38 reference genome with liftOver UCSC tool [99]. Promoters were defined as 2 kb upstream and downstream from transcription start sites. Copy number variation (CNV) was calculated from MeDIP-seq data by considering only fragments without any CpG, based on 2-Mb windows and a fragment size of 250 bp.

Methylation microarray data from 196 CLL patients’ samples (CLLmethylation) covering 435,155 CpGs (we were able to determine hg38 positions for 435,102 CpGs) were obtained via BloodCancerMultiOmics2017 R packages [34] with ExperimentHub and filtered out 53 CpGs without hg38 genomic information. We excluded data from 22 patients without *SF3B1* mutational status information. Differentially methylated probes were called using lmFit and eBayes functions of limma [35] and filtered according to adjusted *p*-value and fold change thresholds applied for the MeDIP-seq data, as described above.

### 4.5. RNA-Seq and Differential Expression Analysis

RNA libraries were generated using the TruSeq Stranded Total RNA sample preparation kit (Illumina, San Diego, CA, USA) which includes a Ribo-Zero depletion of ribosomal RNAs prior to library preparation. Sequencing of 50 bp paired-end reads was performed on a HiSeq2000 (Illumina, San Diego, CA, USA). RNA-seq reads were mapped to the same reference with STAR v2.6.0c [100] and GENCODE gene annotation v36 [101]. Differential expression analysis was performed with edgeR v. 3.30.3 [102]. A gene was considered significantly differentially expressed if the FDR-adjusted *p*-value was < 0.05. To identify possible genes affected by the differential methylation of weak enhancers, no threshold on log2(fold change) was set.

### 4.6. Bisulfite Mass Spectrometry (BS-MS) with Agena Bioscience EpiTYPER-Assay

We selected 16 DMRs with |log2 fold change| > 2 for validation by the EpiTYPER. Primers were designed for CpGs within 16 DMRs. We used a subset of six CLL samples with and six CLL samples without *SF3B1*^mut^.

For the validation of the differentially methylated regions identified by MeDIP-seq, we used the EpiTYPER assay, which is based on bisulfite conversion followed by PCR amplification using one primer containing a T7 promoter sequence, followed by in vitro transcription and Uracil-specific cleavage of the RNA. Fragments were then analyzed by matrix-assisted time-of-flight mass spectrometry (MALDI-TOF) mass array analysis [103,104].

Primers for the EpiTYPER assay were designed using the online tool EpiDesigner with default settings (www.epidesigner.com accessed on 24 March 2021) and purchased from Integrated DNA Technologies (Leuven, Belgium). Oligo sequences, genomic coordinates, and annealing temperatures are given in Appendix A. For the assay, 1 µg of genomic DNA was bisulfite-converted using the EZ DNA Methylation kit (Zymo Research Europe GmbH, Freiburg, Germany) according to the manufacturer’s recommendations. Bisulfite converted DNA was eluted in 60 µL and 1 µL of the dilution was used for amplification in 384-well plates in a 5-µL reaction volume using 0.2 U of HotStarTaq (Qiagen, Hilden, Germany), 1 pmol of each oligo, and 1 nmol of dNTPs. The reaction was run in a thermocycler at 95 °C for 15 min and 35 to 45 cycles at 94 °C for 30 s, annealing (52, 56, or 60 °C) for 30 s, 72 °C for 1 min, and a final extension of 5 min at 72 °C. Subsequently, the PCR product was in vitro transcribed and enzymatically cleaved using the MASSCleave T7 Kit (Agena Bioscience GmbH, Hamburg, Germany) and run on a MassArrayDX (Agena Bioscience GmbH, Hamburg, Germany). A DNA methylation standard was generated by whole genome amplification, WGA using Repli-G (Qiagen, Hilden, Germany) and in vitro methylation using *M.SssI* (CpG) methyltransferase (New England Biolabs, Frankfurt am Main, Germany). Standards of 0%, 20%, 40%, 60%, 80%, and 100% were generated by mixing WGA DNA with WGA and in vitro-methylated DNA. For each assay, a methylation standard was run in parallel. Methylation for individual CpG units was calculated with the EpiTYPER 1.3 software. Subsequently, methylation values (0 to 1) for a given region were calculated as the mean of the analyzed CpG units that passed the quality criteria. CpG units were excluded from the analysis when: (i) less than 50% of all samples had values; (ii) CpG units within one amplicon had an identical mass; and (iii) > 3 CpGs within one CpG unit. DNA methylation values for the amplicons are given in Appendix A.

### 4.7. Gene Set Enrichment Analysis

We collected unique genes that contained at least one significant DMR (FDR < 0.05, |log2 fold change| ≥ 1) within their gene bodies or promoter regions and subjected to functional enrichment analysis with gProfiler2 with default options (version e103_eg50_p15_eadf141) [105,106].

### 4.8. Motif Enrichment Analysis

Differentially methylated 250-bp regions in CLL patients with *SF3B1* mutation compared to CLL without *SF3B1* mutation were used as input for findMotifsGenome.pl module of HOMER v4.11.1 [107] with the following additional options: -nomotif -known -cpg -size 250. The DMRs were analyzed against 440 known motifs identified in vertebrate genomes and available for hg38 annotation provided by HOMER.

### 4.9. Assessing the CLL Subtype

For this part of analysis, we remapped samples to the hg19 reference genome (Genome Reference Consortium—GRCh37) and ran the QSEA R package v.1.14.0 [33], as before. We used mean beta normalized values of the 250 bp bins that overlapped at least 145 bp with the loci defined to classify CLL samples into methylation programming subtypes from Oakes et al. [25]. The merged table with beta values of 27 samples from this study and 329 from the CLL Research Consortium (CRC) in Oakes et al., available in Appendix A, was used to cluster the samples and draw a heat map with the same software (Qlucore, Lund, Sweden, trial version) and settings used by the authors [25].

## Figures and Tables

**Figure 1 ijms-22-09337-f001:**
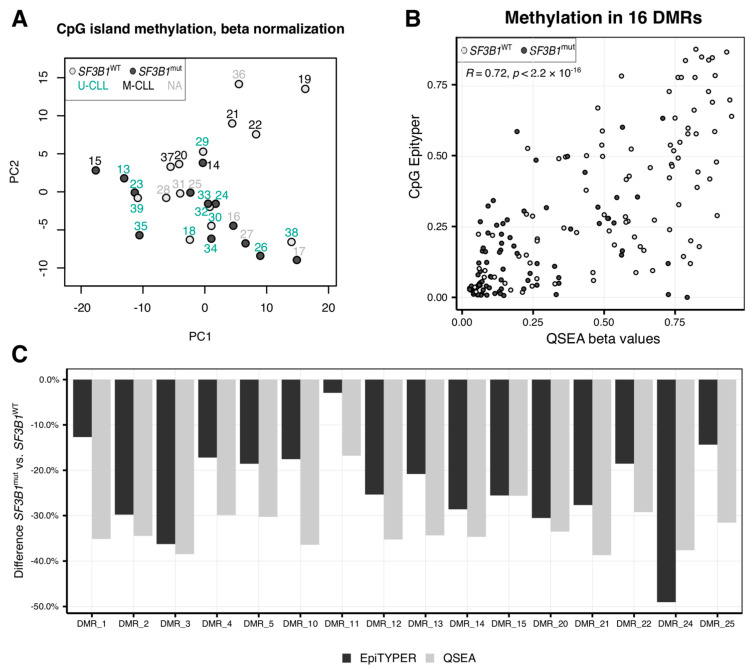
Hypomethylation in chronic lymphocytic leukemia (CLL) patients with *SF3B1* mutation. (**A**) Principal component analysis (PCA) of the samples revealed clustering of the *SF3B1*^mut^ samples with darker points corresponding to the samples with a mutation. PCA was performed using beta normalized values for the 1,000 most variable 250 bp windows within CpG islands. Points colors correspond to the *SF3B1* mutational status. Sample IDs were colored according to *IGHV* mutational status. (**B**,**C**) Validation of 16 from 67 differentially methylated regions (DMRs) identified with the subset of samples (6 *SF3B1*^WT^ and 6 *SF3B1*^mut^) using the EpiTYPER to estimate CpGs methylation levels within the DMRs: (**B**)—correlation between EpiTYPER and beta methylation for each sample and every DMR/CpG; (**C**) comparison of the methylation difference between *SF3B1*^mut^ and *SF3B1*^WT^ CLL mean methylation levels.

**Figure 2 ijms-22-09337-f002:**
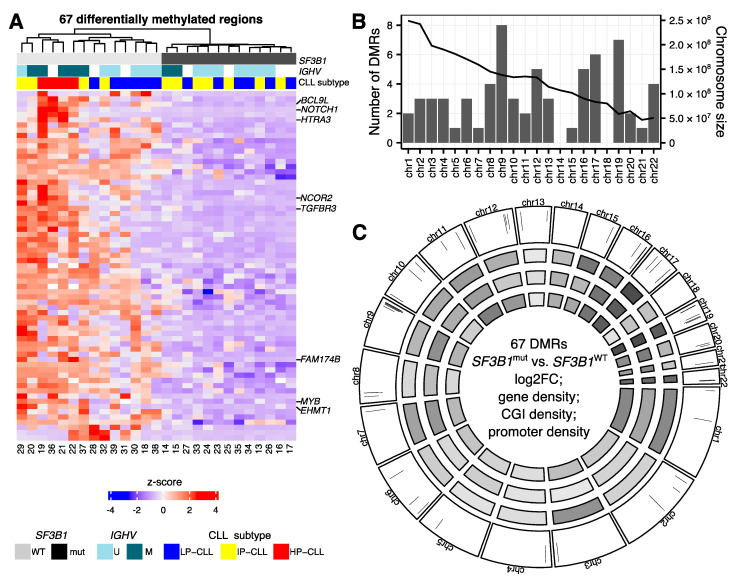
Differentially methylated regions (DMRs) in chronic lymphocytic leukemia (CLL) patients with *SF3B1* mutation. (**A**) Methylation levels of 67 DMRs between CLL patients with and without *SF3B1* mutation. The samples were clustered based on Euclidean distances with a complete linkage agglomeration method. White cells in the *IGHV* annotation denote missing information. CLL-subtype was defined by Oakes et al. based on the 18 loci selected by the authors [25]: low-programmed (LP-CLL), intermediate-programmed (IP-CLL), and high-programmed (HP-CLL). See Figure 4 and Appendix A. (**B**) Numbers of DMRs between CLL patients with and without *SF3B1* mutation identified per chromosome. (**C**) Circos plot with the outer panel showing the log2 fold change in the 67 DMRs between *SF3B1*^mut^ and *SF3B1*^WT^ CLL patients in a −3.7 to 0.2 range. The next three panels show, from outside to inside, the average density of 250 bp regions tested for differential methylation per chromosome within (i) genes; (ii) CpG islands (CGI); (iii) gene promoters. The darker the grey color, the higher the density.

**Figure 3 ijms-22-09337-f003:**
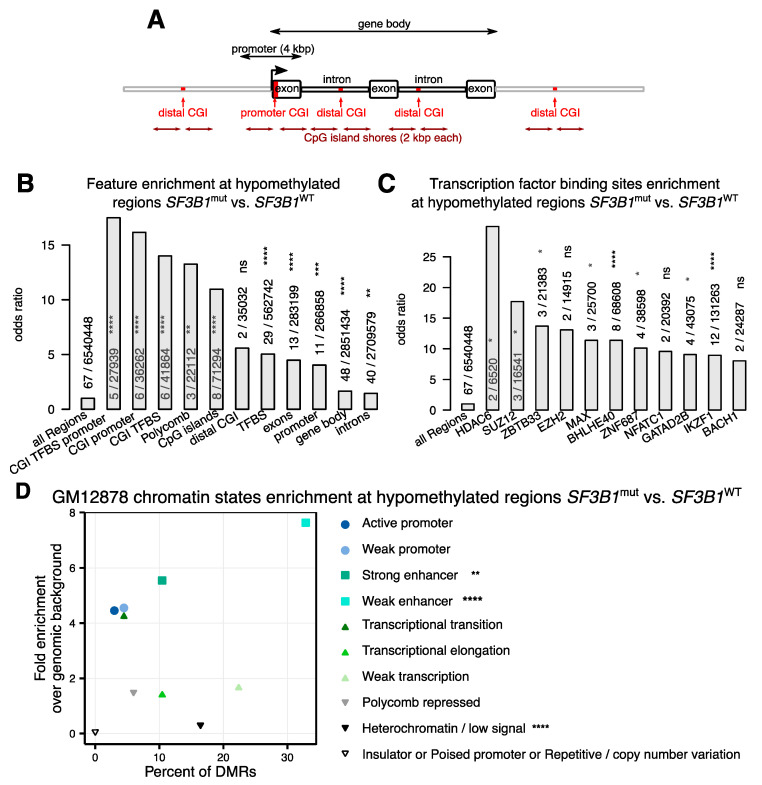
Genomic features of *SF3B1*^mut^ vs. *SF3B1*^WT^ differentially methylated regions (DMRs) at binding sites. (**A**) Schematic visualization of features used in (**B**) A promoter was defined as a region +/− 2000 base pairs (kbp) from a transcription start site. CpG islands (CGI) were obtained from [47]. A CGI shore was defined as a 2-kbp region flanking a CGI up—and downstream. A distal CGI is any CGI outside of promoter regions. (**B**) Enrichment analysis of genomic features with differentially methylated regions (DMRs) shown by odds ratio. Enhancer regions are taken from [48]. Transcription factor binding sites (TFBS, *n* = 122) were derived from ENCODE Encyclopedia v.3. PRC2 binding sites were defined as binding sites for EZH2 or SUZ12. (**C**) Enrichment analysis of the TFBS listed in (**B**)—only 11 TFBSs with the highest odds ratio are shown. (**D**) Enrichment analysis of the chromatin states as derived from GM12878 cell line [49,50]. Significance is denoted by stars with adjusted *p*-value (false discovery rate) < 0.05 = “*”; < 0.01 = “**”; < 0.001 = “***”; < 0.0001 = “****”; ≥ 0.05 = “ns”.

**Figure 4 ijms-22-09337-f004:**
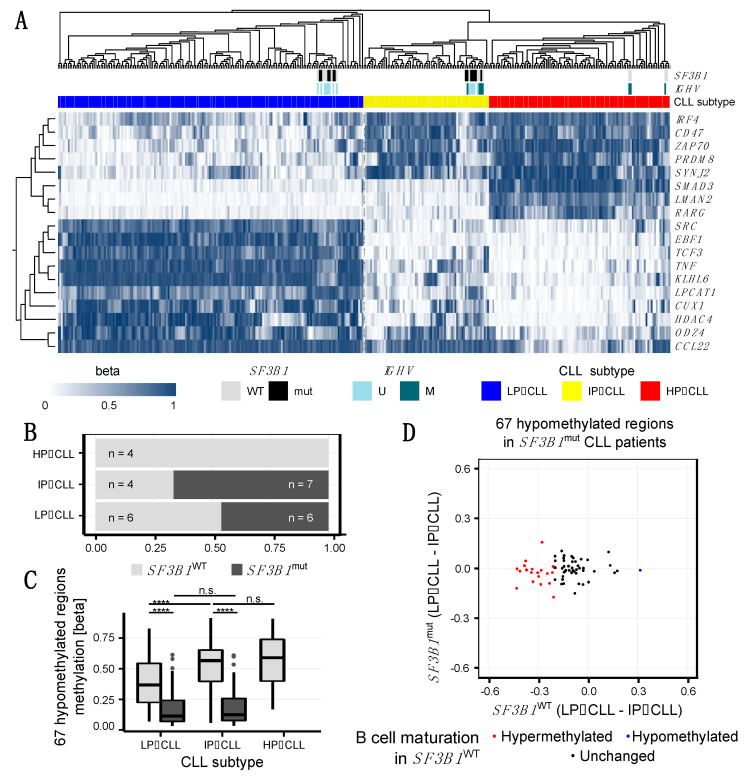
Changes in the DNA methylome in *SF3B1*^mut^ chronic lymphocytic leukemia (CLL) samples occur partly independent of B cell maturation. (**A**) Heatmap showing unsupervised clustering of 27 samples from this study and 329 CLL Research Consortium (CRC) samples from Oakes et al. [25] based on beta methylation values of the 18 most variable regions among the three CLL subtypes. White cells in the *SF3B1* and *IGHV* annotations denote missing information. The inclusion of our 27 samples led to re-classification of 5 out of the 329 CRC samples (<2%) from the LP-CLL to the IP-CLL subtype. (**B**) Proportion of *SF3B1*^WT^ and *SF3B1*^mut^ samples among the three CLL subtypes. (**C**) Methylation values of the 67 hypomethylated regions *SF3B1*^mut^ samples according to the *SF3B1* mutational status and CLL subtype. Wilcoxon test significance is denoted by stars with *p*-value < 0.0001 = “****”; ≥ 0.05 = “ns”. (**D**) Comparison of the methylation programming between CLL patients with and without *SF3B1* mutation within 67 differentially methylated regions (DMRs) identified between *SF3B1*^mut^ CLL and *SF3B1*^WT^ CLL samples. Due to the lack of high-programmed CLL samples with *SF3B1*^mut^ in our dataset (**B**), the methylation programming is shown as the difference between the average methylation of low-programmed (LP-) and the average methylation of intermediate-programmed (IP-) CLL samples in each of the *SF3B1* mutational groups. (**B**) cell development-related DMRs hypo- or hyper-methylated in LP-CLL vs. IP-CLL in wild type samples (difference ≥ 20% in *SF3B1*^WT^) are shown in blue and red, respectively.

## Data Availability

The authors will provide the data upon reasonable request.

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
