# Peer review of "Altered DNA Methylation Profiles in SF3B1 Mutated CLL Patients"

_ijms, 2021, doi:10.3390/ijms22179337_

Round 1
Reviewer 1 Report
- The title is accurate and relevant
- The introduction is sufficient although more references concerning DNA methylation in CLL could be included.
- Tables and Figures are relevant and well presented, although Supplementary Figure 1 could be omitted. Moreover, Table S6 would benefit from editing (modifying the text style).
- The results section could benefit from shortening and references relevant to the association with previously published data would be better to be present at the discussion section.
- Discussion could be improved.
Author Response
We would like to thank reviewer #1 for the kind revision of our work. We tried to address all the following comments in the new version of the manuscript.
- The title is accurate and relevant
We would like to thank the reviewer for this positive statement.
- The introduction is sufficient although more references concerning DNA methylation in CLL could be included.
We would like to thank the reviewer for this comment. In our initial introduction references 16-23 contained articles on methylation changes in CLL in the close context of our study. We now added a few more references (references 16, 20-22 in the subsequent version) in the newer version of the manuscript.
- Tables and Figures are relevant and well presented, although Supplementary Figure 1 could be omitted. Moreover,Table S6 would benefit from editing (modifying the text style).
According to the reviewer’s suggestion we have deleted Figure S1 and improved the font in Table S6.
- The results section could benefit from shortening and references relevant to the association with previously published data would be better to be present at the discussion section.
We have shortened the Results part by moving parts into the Discussion (lines 452-4, 475-9).
- Discussion could be improved.
We would like to thank the reviewer for this comment. We have revised and modified the Discussion by adding a paragraph on the possible impact of the alternative splicing on methylation profile (lines 359-65 and 370-81) as well as by moving parts from the Results part into the Discussion as indicated in the previous comment.
Reviewer 2 Report
In this work, the authors reported that SF3B1 mutations in CLL were associated with hypomethylation of 67 genomic regions, mostly in proximity to telomeric regions and within gene bodies of some cancer-related signaling genes, e.g., NOTCH1, HTRA3 and BCL9L. Some of their results were validated by publicly available dataset. SF3B1 mutations were also enriched in the LP and IP subtypes of CLL. By subgroup analysis, the authors confirmed that the 67 genomic regions were still hypomethylated within each subgroup. They concluded that mutations in SF3B1 could cause additional epigenetic aberrations during CLL development. Although the current analysis is largely descriptive, the authors generated an important dataset for subsequent CLL study. Importantly, this study also shed new light on the previously unreported linkage between SF3B1 and DNA methylation. The manuscript is also well written and organized. From the reviewer’s point of view, the manuscript should be publishable in its present form.
Author Response
We are very grateful for this positive feedback.Reviewer 3 Report
In the manuscript, Pacholewska and colleagues investigate the methylation landscape of CLL depending on the presence of mutations of the SF3B1 gene. Authors analyze the epigenetic profile of 27 CLL patients with methylated DNA immunoprecipitation sequencing and identify 67 loci hypomethylated in SF3B1-mutated cases. Via bioinformatic analyses and comparisons with published datasets, Authors associate SF3B1 mutations with a low-programmed epigenetic state and dissect different DMRs (differentially methylated regions) related or not to normal B cell development, suggesting that SF3B1 mutations cause additional epigenetic aberrations during carcinogenesis.
The manuscript addresses a very interesting topic which has not yet been investigated in CLL but potentially described in myelodysplastic syndromes (Yoshimi et al, Nature 2019); however, several points need to be addressed, as detailed below.
- The main issue is that the data here presented are mostly correlative and a rationale for the reported methylation differences is not clearly presented. Is there any specific SF3B1-induced splicing alteration that may explain at least some of the results? The issue of potential gain-of-function splicing alterations should be at least discussed.
- Does any of the genes in close proximity to DMRs effectively displaying alterations in their expression patterns? Or downstream genes (as in the case of IKAROS?)? Authors should try to perform some Gene Expression Profiling / RNA-seq experiments to functionally validate their observations.
- Have DMRs been evaluated also according concordant/discordant methylation within the region?
- Authors report in Figure 1A and line 114 that no clear cluster is visible; to me seems that most SF3B1-mut cases cluster in the lower part of the graph, with PCA2 mostly resembling a mixture of methylation status and IGHV status, which is strongly associated with SF3B1 mutations (in fact the borderline samples 14-15 cluster on the right side of figure 2A). Since many features of CLL associate with IGHV status, proper control of this confounding factor is crucial. Also lines 329-330 should be re-evaluated.
- Authors should be more complete in describing the methods for mutation detection (TP53, SF3B1, other genes?) as it is crucial for the correct identification of all SF3B1 mutations. What is the detection limit of mutation VAF? And how was the validation with “RNA-sequencing” performed?
- Line 149: a “maximum abs(log2(FC)) = 0.3 within ACOX3 gene” but this value is not present in table s2
- The observation of an involvement of the NOTCH1 pathway seems interesting and an additional clue to the data provided by Wang et al (ref. 35) and Pozzo et al. (Haematologica 2021).
- Lines 243-244: This whole concept is unclear. Why selecting 33% DMRs? Or that was done upon a 20% cut-off? Also description of Figure 4D should be improved since meaning is not obvious.
Clerical errors:
-lines 225-226: extend
- line 471: where
- line 477: to differentially
Author Response
Reviewer #3
In the manuscript, Pacholewska and colleagues investigate the methylation landscape of CLL depending on the presence of mutations of the SF3B1 gene. Authors analyze the epigenetic profile of 27 CLL patients with methylated DNA immunoprecipitation sequencing and identify 67 loci hypomethylated in SF3B1-mutated cases. Via bioinformatic analyses and comparisons with published datasets, Authors associate SF3B1 mutations with a low-programmed epigenetic state and dissect different DMRs (differentially methylated regions) related or not to normal B cell development, suggesting that SF3B1 mutations cause additional epigenetic aberrations during carcinogenesis.
The manuscript addresses a very interesting topic which has not yet been investigated in CLL but potentially described in myelodysplastic syndromes (Yoshimi et al, Nature 2019); however, several points need to be addressed, as detailed below.
We thank reviewer #3 for the thorough revision of our work and constructive criticism.
- The main issue is that the data here presented are mostly correlative and a rationale for the reported methylation differences is not clearly presented. Is there any specific SF3B1-induced splicing alteration that may explain at least some of the results? The issue of potential gain-of-function splicing alterations should be at least discussed.
We thank the reviewer for this comment which we have discussed internally several times, but had decided not to include in the previous version of the manuscript. We have now addressed this point in the new version of the manuscript. We have compared our identified DMRs with data of publicly available data on splicing alterations upon SF3B1 mutations (Wang et al.) and have observed EHMT1and UCKL1 to be differentially methylated and alternatively spliced in SF3B1 mutated samples. We have included this finding in the Discussion section (lines 370-81). Thus far no splicing alteration has been identified in any of the DNA methyl transferase (DNMTs) genes in CLL patients with SF3B1mut. Interestingly, Yoshimi et al. have shown a synergistic effect on the methylation profile between Idh2mutand Srsf2mut in acute myeloid leukemia. Furthermore, based on our experiments, we could not associate the methylation changes with specific alteration in splicing. We added these observation in lines 359-65.
- Does any of the genes in close proximity to DMRs effectively displaying alterations in their expression patterns? Or downstream genes (as in the case of IKAROS?)? Authors should try to perform some Gene Expression Profiling / RNA-seq experiments to functionally validate their observations.
We are very thankful for this comment. As far as gene expression is concerned, (also lines 382-384 of the initial manuscript), only two genes that contained overlapping DMRs (gene body or promoter region) showed increased expression based on the RNA-seq experiment. Furthermore, the DMRs were significantly enriched in enhancer regions (n = 29), which makes it more difficult to associate them with the gene expression due to the fact that long distance (or even inter-chromosomal) promoter-enhancer interactions exist. Nevertheless, we decided to provide additional information about the distance between enhancer regions and nearest genes in Table S2. In particular, it now contains the information on the distance between the DMR to the closest promoter and to the closest promoter of a differentially expressed gene (columns AK-AP). According to Chepelev et al. 2012 only 100 out of 2,373 enhancer-promoter interactions were observed at the range above 1Mb. In our data, from 29 DMRs in the enhancer regions, 21 (72%) were within 1Mb to a significantly differentially expressed gene’s promoter. This information was added in lines 212-8 and the RNA-seq experiment was also added to the Methods part (lines 577-85).
- Have DMRs been evaluated also according concordant/discordant methylation within the region?
We would like to thank the reviewer for this interesting point. However, since the technology used is based on an antibody to enrich for methylated DNA fragments and thus does not assess individual CpGs, we are not able to thoroughly investigate the reviewer’s question. Nevertheless, in our EpiTYPER validation experiments, the DNA methylation level for single CpG sites can be assessed for 16 DMRs. For two of the 16 DMRs methylation only a single CpG per DMR could be assessed with the EpiTYPER. We have provided these data in Table S4. We added the cell color according to the methylation values. Although the EpiTYPER data cannot provide methylation information for individual CpGs localized on the same DNA strand, the bulk methylation level of neighboring CpG sites reveals that the methylation of the CpGs within the 13 DMRs seems mostly concordant. We added this information to the manuscript text (lines 137-40).
- Authors report in Figure 1A and line 114 that no clear cluster is visible; to me seems that most SF3B1-mut cases cluster in the lower part of the graph, with PCA2 mostly resembling a mixture of methylation status and IGHV status, which is strongly associated with SF3B1 mutations (in fact the borderline samples 14-15 cluster on the right side of figure 2A). Since many features of CLL associate with IGHV status, proper control of this confounding factor is crucial. Also lines 329-330 should be re-evaluated.
We agree with the reviewer that there may be two clusters with the SF3B1mut samples concentrating in the lower part, and SF3B1WT patients in the upper part of the PCA plot. However, due to the lack of a clear separation of the two groups we chose not to state these clusters were indeed separating the two groups, especially due to WT samples 18, 30, 38 appearing also in the lower part. Instead, we chose to say that although some separation could also be observed based on the SF3B1 mutational status, no clear clusters grouped by the SF3B1 genotype were visible (lines 127-30). Figures 1A, 2A, 4A and S6 have been updated.
We also agree with the reviewer that the IGHV status and methylation programming status, which have been previously reported to be associated with SF3B1mut patients (Vollbrecht et al, 2015 and Wojdacz et al. Blood Adv. 2019, Bhoi et al. Epigenetics 2016, Queirós et al. Leukemia 2015), are an important aspect of our study. It is possible that the IGHV status has an even stronger effect on the methylation profile than SF3B1 mutations, as already visible in PCA, which is based on 1,000 most variable CpG regions. However, the methylation at the 67 identified DMRs seems to be higher in all SF3B1WT CLL patient compared to SF3B1mut patients, regardless of the IGHV status, as shown in Figure 2A.
We did observe and discussed thoroughly the overlap between the methylation affected by the methylation programming due to normal B-cell differentiation (and therefore, the IGHV mutational status), but we did also observe some changes that seem to be independent of this process. This finding was very interesting for us and was the major focus of our study. We tried to be very open and clearly stated that some of the DMRs may be due to the differential status of the originating B-cell (and therefore IGHV status). As the reviewer suggested, we modified lines 384-5 (now lines 383-4).
- Authors should be more complete in describing the methods for mutation detection (TP53, SF3B1, other genes?) as it is crucial for the correct identification of all SF3B1 mutations. What is the detection limit of mutation VAF? And how was the validation with “RNA-sequencing” performed?
We absolutely agree with the reviewer that this information is important. A detailed description of the method for the detection of the mutations has been included in a previous publication. We have added the information in the manuscript in the Materials and Methods section paragraph 4.1, in lines 510-3, and expanded Table S1 with information on more genes. In particular for TP53 and SF3B1 the complete coding regions have been analyzed by a multiplex PCR-panel followed by next generation sequencing.
- Line 149: a “maximum abs(log2(FC)) = 0.3 within ACOX3 gene” but this value is not present in table s2
We would like to thank the reviewer for this comment. The log2FC was indicated in column AI of the Table S2. We added the column ID to the manuscript to clarify it (line 166).
- The observation of an involvement of the NOTCH1 pathway seems interesting and an additional clue to the data provided by Wang et al (ref. 35) and Pozzo et al. (Haematologica 2021).
Yes, that is a very interesting connection. We improved the manuscript text in line 398-9 and included the publication by Pozzo et al.
- Lines 243-244: This whole concept is unclear. Why selecting 33% DMRs? Or that was done upon a 20% cut-off? Also description of Figure 4D should be improved since meaning is not obvious.
We would like to thank the reviewer for this comment. We did not select the 33% DMRs – this is the part of the DMRs identified initially (SF3B1mut vs. SF3B1WT) for which we also observed the change between LP- and IP-CLL in WT samples. We edited the manuscript text in lines 272-6 and the Figure 4D description.
Clerical errors:
-lines 225-226: extend
Thank you, we have extended the text in lines 242-3.
- line 471: where
Thank you, we have corrected this mistake.
- line 477: to differentially
Thank you, we have corrected this mistake.